# A T-Cell-Derived 3-Gene Signature Distinguishes SARS-CoV-2 from Common Respiratory Viruses

**DOI:** 10.3390/v16071029

**Published:** 2024-06-26

**Authors:** Yang Li, Xinya Tao, Sheng Ye, Qianchen Tai, Yu-Ang You, Xinting Huang, Mifang Liang, Kai Wang, Haiyan Wen, Chong You, Yan Zhang, Xiaohua Zhou

**Affiliations:** 1Beijing International Center for Mathematical Research, Peking University, Beijing 100871, China; yang.li@cqbdri.pku.edu.cn; 2Chongqing Research Institute of Big Data, Peking University, Chongqing 400041, China; xinya.tao@cqbdri.pku.edu.cn (X.T.); xinting.huang@cqbdri.pku.edu.cn (X.H.); 3Chongqing Center for Disease Control and Prevention, Chongqing 400707, China; yshscu@163.com; 4Department of Probability and Statistics, School of Mathematical Sciences, Peking University, Beijing 100091, China; 2101110081@pku.edu.cn; 5Institute of Pharmaceutical Science, King’s College London, London WC2R 2LS, UK; yuang.you@kcl.ac.uk; 6NHC Key Laboratory of Medical Virology and Viral Diseases, National Institute for Viral Disease Control and Prevention, Chinese Center for Disease Control and Prevention, Beijing 102206, China; mifangl@163.com; 7Key Laboratory of Molecular Biology for Infectious Diseases (Ministry of Education), Department of Infectious Diseases, Institute for Viral Hepatitis, The Second Affiliated Hospital of Chongqing Medical University, Chongqing 400010, China; wangkai@cqmu.edu.cn; 8Chongqing International Travel Health Care Center, Chongqing 401120, China; wenhaiyan2001@163.com; 9Shanghai Institute for Mathematics and Interdisciplinary Sciences, Fudan University, Shanghai 200433, China; 10Sports & Medicine Integration Research Center (SMIRC), Capital University of Physical Education and Sports, Beijing 100088, China

**Keywords:** biomarkers, SARS-CoV-2, diagnostics, gene expression, database

## Abstract

Research on the host responses to respiratory viruses could help develop effective interventions and therapies against the current and future pandemics from the host perspective. To explore the pathogenesis that distinguishes SARS-CoV-2 infections from other respiratory viruses, we performed a multi-cohort analysis with integrated bioinformatics and machine learning. We collected 3730 blood samples from both asymptomatic and symptomatic individuals infected with SARS-CoV-2, seasonal human coronavirus (sHCoVs), influenza virus (IFV), respiratory syncytial virus (RSV), or human rhinovirus (HRV) across 15 cohorts. First, we identified an enhanced cellular immune response but limited interferon activities in SARS-CoV-2 infection, especially in asymptomatic cases. Second, we identified a SARS-CoV-2-specific 3-gene signature (CLSPN, RBBP6, CCDC91) that was predominantly expressed by T cells, could distinguish SARS-CoV-2 infection, including Omicron, from other common respiratory viruses regardless of symptoms, and was predictive of SARS-CoV-2 infection before detectable viral RNA on RT-PCR testing in a longitude follow-up study. Thereafter, a user-friendly online tool, based on datasets collected here, was developed for querying a gene of interest across multiple viral infections. Our results not only identify a unique host response to the viral pathogenesis in SARS-CoV-2 but also provide insights into developing effective tools against viral pandemics from the host perspective.

## 1. Introduction

Severe acute respiratory syndrome coronavirus 2 (SARS-CoV-2) infection can result in a range of clinical manifestations, from asymptomatic or mild infection to severe coronavirus disease 2019 (COVID-19), that necessitate intensive care. During the current wave of Omicron, the proportion of asymptomatic cases has increased compared with previous SARS-CoV-2 variants [1]. Asymptomatic persons are defined as individuals who test positive for viral RT-PCR but present no symptoms. While some individuals may go through the entire course of infection and never experience symptoms, others who initially present as asymptomatic may go on to develop symptoms days or weeks later. The individuals who will later develop symptoms are defined as being pre-symptomatic [2]. Of note, these individuals with SARS-CoV-2 who shed viable virus while asymptomatic or pre-symptomatic have been reported to be a major driver of the COVID-19 pandemic, leading to more than 50% transmission cases [3,4,5,6].

Respiratory viruses are a common cause of illness worldwide, with influenza virus (IFV), respiratory syncytial virus (RSV), and human rhinovirus (HRV) being among the most prevalent [7]. These viral infections generally cause mild respiratory illnesses like the common cold but can also progress to fatal acute respiratory distress syndrome, particularly in vulnerable populations such as the elderly and those with underlying health conditions [8]. Emerging evidence suggests the determining factors of the symptoms of infections are more possibly related to host responses than viruses.

In general, the host’s innate immune responses play an essential role in suppressing the replication of the virus once it enters the host, such as antiviral-mediated interferons and cytokines. A study on the transcriptional response of human cell lines to SARS-CoV-2 and common respiratory viruses indicated that impaired interferon (IFN) activity and overproduction of inflammatory cytokines are the driving features of COVID-19 [9]. On the other hand, higher levels of IFN responses were associated with severe disease outcomes in COVID-19 patients [10,11]. This dichotomy underscores the understanding of the unique pathogenesis of COVID-19.

Measurement of the host response genes, as opposed to viral targets, is one of the most promising diagnostic strategies. A range of blood signatures has been proposed as candidate diagnostic biomarkers for various purposes, including bacterial/viral discrimination [12,13,14], diagnosis of pre-symptomatic viral infection in known contacts [15], or diagnosis of specific viral infections, such as IFV [16]. These studies shed light on uncovering the host genes that are specific to SARS-CoV-2 infections, regardless of symptoms.

Here, we collected transcriptome profiles from 3730 samples across 15 independent cohorts. Next, we conducted integrative analyses utilizing bioinformatics and machine learning approaches on both asymptomatic and symptomatic individuals infected with SARS-CoV-2 and prevalent respiratory viruses. We identified a SARS-CoV-2-specific host response gene signature that was sourced from T cells. Moreover, this signature distinguished SARS-CoV-2 samples from healthy controls or non-SARS-CoV-2 viral infections with high accuracy.

## 2. Materials and Methods

### 2.1. Dataset Collection

The overall design of this study is displayed in Figure 1. We conducted a thorough search of public data repositories related to respiratory viral infections, including SARS-CoV-2, Influenzas Virus (IFV), Human Rhinovirus (HRV), Respiratory Syncytial Virus (RSV), and seasonal Human Coronaviruses (sHCoVs). We excluded datasets that (i) were nonclinical, (ii) were profiled using tissues other than WB or PBMCs, (iii) did not have samples from at least 3 healthy controls, or (iv) did not provide information to identify whether a patient had a bacterial or viral infection.

Subsequently, we curated 15 independent cohorts, comprising 3730 biological samples from the Gene Expression Omnibus (GEO), Sequence Read Archive (SRA), ArrayExpress, and Genome Sequence Archive (GSA). Overall, the RNA profiles in these datasets were extracted from whole blood or peripheral blood mononuclear cells (PBMCs) and covered a diverse range of children and adults. We categorized the samples based on their biological conditions, distinguishing between viral infections and healthy controls using established techniques for virus detection such as viral culture and polymerase chain reaction (PCR), as detailed in the original publications (Appendix A).

The discovery train sets included asymptomatic and symptomatic cases with SARS-CoV-2 and common respiratory virus infections. The cohort of individuals with SARS-CoV-2 infection in Shanghai, China, cohort consisted of blood samples from 15 healthy donors, 16 asymptomatic and pre-symptomatic cases who were positive for SARS-CoV-2 tests but manifested no symptoms, and 16 COVID-19 patients at acute phase (7 mild and 9 moderate) during March 2020 to November 2020. It should be noted that RNA sequencing was obtained before treatment. Raw FASTQ files and clinical information were achieved through accession number HRA000786 from the Genome Sequence Archive (GSA) website (http://bigd.big.ac.cn/gsa-human, accessed on 23 July 2023) [17]. In GSE17156, Zaas and colleagues [18] inoculated healthy volunteers with one of the following viruses (HRV, RSV, and IFV). The overall attack rates were around 50% (50% for HRV, 45% for RSV, and 53% for IFV). Three cohorts, focusing on SARS-CoV-2 and comprising both hospitalized patients and healthy controls, served as the test sets in the discovery group. Within GSE161918 [19], only samples from COVID-19 cases applied near hospital admission, paired with age- and sex-matched healthy donors (19 symptomatic, 14 controls), were included and retained. In GSE171110 [20] and GSE152641 [21], samples from patients were also collected at or near the time of hospital admission.

During the validation stage, we scrutinized the diagnostic ability of the biomarkers for SARS-CoV-2 infection across five external COVID-19 cohorts. Specifically, GSE166190 [22] was selected to evaluate diagnostic performance in symptomatic adults and children with SARS-CoV-2 infections, providing insights into age-specific outcomes. We also examined the diagnostic performance in cohorts GSE198449 [23] and GSE157103 [24], which presented data for individuals with various clinical conditions, including subjects from healthy populations and those with acute respiratory distress syndrome (ARDS). Notably, we identified a cohort (GSE201530) containing asymptomatic and symptomatic cases infected with Omicron [25], allowing us to assess the robust performance of biomarkers for SARS-CoV-2 variants. The expression data and detailed clinical information of E-MTAB-10022 were archived from ArrayExpress [26]. One key feature of E-MTAB-10022 was the weekly longitudinal follow-up of study participants, which enabled systematic evaluations of blood transcriptional biomarkers before and at the point of SARS-CoV-2 PCR positivity.

Additionally, we selected five datasets featuring cases of IFV, HRV, RSV, and sHCoVs to assess the specificity of the biomarkers. The GSE38900 datasets [27] focused on children with lower respiratory tract infections, encompassing IFV, RSV, and HRV cases. The GSE68310 [28], containing samples presenting acute respiratory illnesses caused by various viruses, including IFV, HRV, and sHCoVs, was also retrieved. GSE105450 [29] included a prospective observational cohort study involving a convenience sample of previously healthy children under two years of age with acute RSV infection and healthy non-infected age-matched controls. GSE61754 [30] presented data from an influenza challenge study in which 22 healthy adults (11 vaccinated) were inoculated with H3N2 influenza (A/Wisconsin/67/2005). In GSE67059, Heinonen et al. [31] conducted a prospective enrollment of children under two years old with an HRV infection history over six respiratory seasons from four study medical centers and evaluated the value of gene expression profiles.

### 2.2. Data Preprocessing

We renormalized all microarray datasets using standard methods when raw data were available from the GEO. We applied robust multiarray average (RMA) to arrays with mismatch probes for Affymetrix arrays. We used normal-exponential background correction and quantile normalization in the *lumi* package for Illumina and Agilent arrays. Thereafter, the probe sets with known gene symbols were kept for downstream analysis.

For RNA-seq cohorts, we obtained the raw reads of the SARS-CoV-2 cohorts if raw reads were available. First, the raw reads were subject to quality control, which involves trimming low-quality and adapter reads, removing low-complexity sequences, and retaining reads of trimmed length >100 bp using Fastp (v0.23.2) [32]. Second, we processed the clean reads with Salmon (v1.9.0) [33] against human transcriptome sequences from the GENCODE site (v38) to obtain the gene-level count matrix. Finally, the count matrix was normalized for further analysis by variance Stabilizing Transformation (vst) function in R package DESeq2 [34].

### 2.3. Re-Standardized Group

For each dataset, we used the sample phenotypes as defined in the original publication. We manually assigned a re-defined severity category to each sample based on the cohort description for each dataset in the original publication, as follows: (1) Controls—uninfected healthy individuals; (2) asymptomatic—individuals who tested positive for a virus but presented no symptoms, including asymptomatic and pre-symptomatic cases; and (3) symptomatic—symptomatic individuals with the definite viral infection.

### 2.4. Differentially Expressed Genes Screening

To screen the viral infections associated with differentially expressed genes (DEGs), DEGs analyses contrasting the virus-infected asymptomatic and symptomatic samples and the healthy samples were performed, respectively. For microarray, DEGs were screened by function for linear model fitting in the R package limma (v3.58.1) [35]. For RNA-seq, DEGs were retrieved by R package DESeq2 (v.1.42.0) [34]. Correction for multiple testing was addressed by controlling the false discovery rate (FDR) using the Benjamini and Hochberg (B.H.) method. Criteria for DEGs were an absolute log2 fold change (Log2FC) of 0 and an FDR-adjusted *p*-value of < 0.05.

### 2.5. Gene Ontology (GO) and Kyoto Encyclopedia of Genes and Genomes (KEGG) Analyses

To understand the functions of enriched genes in interesting modules, GO and KEGG analyses were performed using clusterProfiler (v4.10.0) [36], identifying significant results based on an FDR-adjusted *p*-value < 0.05.

### 2.6. Multiple-Cohort Integration and Co-normalization

As shown in previous studies [37], we applied Combat CONormalization Using conTrols (COCONUT) into the discovery set (HRA000786 and GSE17156) for between-cohort integration [38]. COCONUT (v1.0.2) allows for co-normalization of gene expression data without bias toward sample diagnosis by applying a modified version of the ComBat empirical Bayes normalization method, which assumes a similar distribution between control samples [39].

### 2.7. Co-Expression Network Construction

A co-expression network was constructed using the integrated discovery sets by weighted correlation network analysis (WGCNA) (v.1.72.5) [40]. Briefly, quality assessment was conducted using the cluster method. The soft-thresholding power was then calculated, with the type of network set to sign. The correlation coefficient threshold was 0.85. Network construction was then performed based on the calculated power. In addition, the minimum number of genes in each module was 30, and the threshold for cut height was set to 0.25 to merge possible similar modules. Finally, the correlation between modules and disease severity was shown by the heatmap.

### 2.8. Estimation of Immune Cell Type Abundances

CIBERSORTx is a machine-learning method that enables the estimation of cell type abundances from bulk transcriptomes [41]. Integrated gene expression data from the discovery datasets were used to characterize the immune cell composition with CIBERSORTx (http://cibersortx.stanford.edu/, accessed on 15 August 2023) with the default signature matrix at 1000 permutations. The sum of all estimated immune cell type fractions was equal to 1 for each sample.

### 2.9. Gene Set Variation Analysis (GSVA) with Pre-Defined Interferon Gene Sets

GSVA is a non-parametric, unsupervised method for estimating the variation of pre-defined gene sets in case and control samples of gene expression data [42]. The pre-defined gene sets in this study were based on the genes induced by the in vitro stimulation of normal human PBMC with interferons (IFN), including IFNA2, IFNB1, IFNW1, and IFNG and referenced with TNF [43] (Appendix A).

### 2.10. Feature Selection with Bioinformatics and Machine Learning

(1) The genes in WGCNA modules that were most related to SARS-CoV-2 were extracted. DEG screening was also performed on all discovery test datasets, including GSE152641, GSE161918, and GSE171110. Genes that appeared in both analyses were considered stable SARS-CoV-2 specific candidate genes.

(2) The SARS-CoV-2 signature discovery was performed in the discovery train dataset. Machine learning algorithms possessed the ability of feature selection, such as random forest, XGBoost, generalized boosted regression modeling (GBM), least absolute shrinkage and selection operator (LASSO), and elastic network were performed on the stable SARS-CoV-2 specific candidate genes. The parameter tuning details were described in the Appendix A. Genes with importance retrieved by each method were aggregated by Robust Rank Aggregation (RRA) [44].

(3) The selected genes were subject to further model building. To maximize the AUC, we adopt the method by John Q. Su and Jun S. Liu [45], which provides the optimal linear combination of multiple diagnostic markers. Briefly, the coefficients for the best linear combination are $w\propto {\left(\Sigma_{x}+\Sigma_{y}\right)}^{-1}$($\mu_{y}$-$\mu_{x}$), which can be estimated by the following:(1)$${\left(\frac{S_{x}}{m-1}+\frac{S_{y}}{n-1}\right)}^{-1}\left(\bar{Y}-\bar{X}\right)$$
where $S_{x}=\sum_{i=1}^{m}(X_{i}-\bar{X}){\left(X_{i}-\bar{X}\right)}^{T}$ and $S_{y}=\sum_{j=1}^{n}(Y_{j}-\bar{Y}){\left(Y_{j}-\bar{Y}\right)}^{T}$.

(4) The locked model was then applied to the validation datasets.

### 2.11. External Validation

We evaluate the classification performance of the SARS-CoV-2 specific signature, along with published signatures containing genes. Briefly, AUC was calculated using “pROC” [46] with the score of the model in the case group being larger than that in the control group. Thereafter, the receiver operating characteristic (ROC) curve was plotted to evaluate the diagnostic performance of the selected genes regarding distinguishing asymptomatic and symptomatic cases with SARS-CoV-2 from all other conditions.

### 2.12. Single-Cell RNA-Seq Analysis

We downloaded the scRNA-seq data for CNP0001102 [47]. We performed quality control and processed both datasets separately with Seurat [48]. After normalizing read counts, we performed principal component analysis (PCA), uniform manifold approximation and projection (UMAP), and shared nearest neighbors clustering on the data. Cell type annotation of clusters was performed with manual annotation using cell type markers as previously described.

### 2.13. Statistical Analysis

R (version 4.1.3) was used for most analyses. Mann–Whitney U test was applied to non-normally distributed variables and the t-test was applied to normally distributed variables. *p*-value < 0.05 was considered statistically significant.

## 3. Results

### 3.1. Clinical Characteristics of the Included Population

We searched the public repositories for blood transcriptome profiles from patients with SARS-CoV-2, HRV, RSV, IFV, and sHCoVs infections. A total of 15 cohorts composed of 3730 samples from patients across continents of Asia, Europe, and North America were archived (Appendix A). Overall, these datasets included a broad spectrum of biological, clinical, and technical heterogeneity represented by blood samples profiled from children and adults infected with a single virus above using either microarray or RNA sequencing. We assigned a re-standardized category to each of the 3730 samples. Briefly, we divided asymptomatic and pre-symptomatic case samples into ‘‘asymptomatic’’, and symptomatic patients into ‘‘symptomatic’’ categories based on original publications. We also defined all non-infected, healthy individuals as “Controls”.

### 3.2. Host Response Profiling in SARS-CoV-2 and Other Common Respiratory Viral Infections

To compare the host response among symptomatic and asymptomatic cases of SARS-CoV-2 with other common respiratory viruses (HRV, IFV, and RSV), we first performed a differential expression analysis, comparing infected individuals with matched healthy controls from original publications (GSE17156 and HRA000786) (Figure 2A). For infected cases presenting symptoms, HRV, IFV, and RSV infections shared similar differentially expressed genes (DEGs), while SARS-CoV-2 induced a set of distinct DEGs (Figure 2A,B). Among the DEGs, the antiviral interferon-stimulated genes (ISGs) were the most up-regulated DEGs found in HRV, IFV, and RSV infections, e.g., IFI44L, IFI27, ISG15, and RSAD2 et al. Although ISGs were observed in SARS-CoV-2 infections (i.e., IFI27), chemokine-related genes, including CXCL2 and CXCL8, were the most upregulated. For asymptomatic infections, HRV, IFV, and RSV induced a distinct pattern of the host response to symptomatic infections. Interestingly, SARS-CoV-2 shared some DEGs in both symptomatic and asymptomatic cases, e.g., IFI27, IL1A, CXCL2, and G0S2.

Next, we performed enrichment analysis over the DEGs results in virus infections using Gene Ontology (GO) terms (Figure 2C). DEGs are involved in multiple biological processes, such as defense response to the virus, interferon response, T cell activation, and cell differentiation, etc. Although ISGs were upregulated in SARS-CoV-2 infections, the interferon (IFN) signaling-related terms (e.g., response to type I IFN, response to type II IFN, etc.) were insignificantly enriched in SARS-CoV-2 infections, regardless of symptoms. In contrast, significant enrichments of IFN responses were found in the other studied viruses. This was in line with previous studies that impaired but not dismissed IFN activities were uncovered in COVID-19 patients [49]. Intriguingly, for the other studied viruses here, the activation of T cells was enriched with significance in symptomatic, other than asymptomatic cases. However, significant enrichments of T cell activation were observed in both symptomatic and asymptomatic individuals infected with SARS-CoV-2. One prominent feature of SARS-CoV-2 infection is lymphopenia, which could augment T cell activation [50].

With the help of the CIBERSORTx, we investigated the differences in immune cell components between subjects with or without virus infections (Figure 3). Individuals with SARS-CoV-2 infection exhibited significant changes in proportions of memory B cells, activated dendritic cells (DCs), monocytes, resting natural killer (NK) cells, CD4+ Naïve T cells, and CD8 T cells (Figure 3A). It should be noted that significant reductions in monocytes and enhancements in CD4+ Naïve T cells were observed in SARS-CoV-2 infection, which showed similar trends in results from flow cytometry [17].

To understand the relative contributions of different responses to IFN between SARS-CoV-2 and other respiratory viral infections, GSVA using IFNA2, IFNB1, IFNW1, IFNG, and TNF signature genes (Appendix A) was employed to calculate the enrichment scores (Figure 3B). The GSVA scores demonstrated cases with HRV, IFV, and RSV, other than SARS-CoV-2, highly enriched signatures for IFNA2, IFNB1, IFNW1, IFNG, and the IFN core, suggesting the inhibition of IFN was induced by SARS-CoV-2. Noting that TNF signatures were exceedingly enriched in SARS-CoV-2 infection, surprisingly in asymptomatic cases.

### 3.3. Identification of SARS-CoV-2 Specific 3-Gene Signature via Integrative Bioinformatics and Machine Learning Approaches

The unique host response to SARS-CoV-2 infection provided the basics to discover SARS-CoV-2-specific feature genes. Firstly, we sought to define the gene module that was correlated with the SARS-CoV-2 infection through weighted correlation network analysis (WGCNA). First, the two datasets HRA000786 and GSE17156 were co-normalized into a single discovery train dataset as described previously [37]. Briefly, we applied the variance stabilizing transformation from DESeq2 (v1.26.0) [34] to normalize gene expression from HRA000786 and the robust multiarray average to each virus challenge study in GSE17156. The batch effects were removed through COCONUT (Appendix A). Second, to ensure that a scale-free network was constructed, a soft-thresholding power of 16 was selected, while 0.85 was regarded as the correlation coefficient threshold (Appendix A). A total of 16 modules were identified (Figure 4A). All color modules were subject to the module–trait correlation analyses. The Pearson correlation analysis, which involved calculating the student *p*-values for the correlations between the MEs of each module and severity, is shown in Figure 5A. The blue module, including 1828 genes, was most relevant to SARS-CoV-2 infection regardless of symptom (Figure 4B).

Next, we extracted genes from the blue module and performed enrichment analyses (Figure 4C,D). The enriched GO biological process terms with viruses focused on the viral life cycle, suggesting the active process of the virus was observed (Figure 4C). Interestingly, the KEGG results suggested the most enriched term was the herpes simplex virus 1 infection, followed by the T cell receptor signaling pathway (Appendix A). In addition to virus-related terms, the GO BP terms include predominately processes related to the cell cycle like RNA splicing, chromosome segregation, DNA replication, etc. The cellular component (CC) results suggested these genes were involved in chromosome division, which was of great importance in the cell cycle. The top three terms for molecular function (MF) were ATPase activity, protein serine/threonine kinase activity, and tubulin binding (Figure 4D).

A total of 1036 genes showed up in both the WGCNA blue module and DEGs from discovery test datasets (Appendix A). These genes were not correlated with HRV, IFV, and RSV (Appendix A). To further reduce to a subset of genes that is more amenable to translation to a point-of-care clinical test, five machine learning approaches with the ability of feature selection were carried out over the 1036 genes in the discovery train dataset. The genes selected by each method were used in RRA. The genes with *p*-values less than 0.05, including CLSPN, RBBP6, and CCDC91, were discovered as the SARS-CoV-2 specific genes (Figure 4E).

To maximize the AUC, we denote by $X_{i}={\left({CLSPN}_{i},\;{{RBBP6}_{i},\;CCDC91}_{i}\right)}^{T}$, i=1,…,m the blood transcript signature values of m patients in the control group and denote by $Y_{j}={\left({CLSPN}_{i},\;{{RBBP6}_{i},\;CCDC91}_{i}\right)}^{T}$, j=1,…,n those of n patients in the case group. Suppose $X_{1},\ldots ,X_{m}$ are independent and identically distributed samples from $N(\mu_{x},\Sigma_{x})$ and $Y_{1},\ldots ,e$ are independent and identically distributed samples from $N(\mu_{y},\Sigma_{y})$. The coefficients could be estimated according to formula (1). After adjusting these coefficients so that their sum is 1, we get the risk score for SARS-CoV-2 infections = 0.53 * CLSPN + 0.37 * RBBP6 + 0.1 * CCDC91. Thereafter, the locked model of 3-gene signature achieved AUC reaching 0.95 to 1 in the discovery test datasets (Figure 4F).

### 3.4. T Cells Are the Primary Source of the 3-Gene Signature

We used single-cell RNA sequencing (scRNA-seq) profiles of 15,639 cells from PBMC samples of 7 individuals (4 SARS-CoV-2 infections, 3 Controls) to identify the cell types that express the 3-gene signature. Visualization using UMAP illustrated the infection status (Figure 5A) followed by cell type (Figure 5B). T cells had the highest scores (Figure 5C), and 3-gene signature scores were significantly higher in T cells from patients with SARS-CoV-2 infection (Figure 5D), especially in activated CD4 T cells, mucosal-associated invariant T cells (MAIT) and Cycling T cells. These results provided further evidence that T cells could be the primary source of the host response at the site of SARS-CoV-2 infection.

### 3.5. Validation of a 3-Gene Signature for SARS-CoV-2: Specificity and Predictive Value

We validated that the 3-gene signature in both discovery and validation datasets were not co-normalized with each other. As expected, sample level 3-gene signature scores for patients with SARS-CoV-2 were higher than those for controls and non-SARS-CoV-2 infections, suggesting the 3-gene signature had not only superior diagnostic power (pooled AUC > 0.9) but also specificity in SARS-CoV-2 infection (Figure 6A). Firstly, four SARS-CoV-2 validation datasets including more than 1800 samples were tested. In GSE157103, where non-SARS-CoV-2 but hospitalized samples were present, 3-gene signature still identify SARS-CoV-2 infections. Although samples from SARS-CoV-2 Omicron patients were not used to discover the 3-gene signature, the 3-gene signature had an AUC of 0.992 (95% CI 0.976–1) in GSE201530. Secondly, cohorts with common respiratory viruses, i.e., IFV, HRV, RSV, and sHCoV, were also investigated. The 3-gene signature had a summary AUC less than 0.6, suggesting no cross-reactions were observed. Next, we investigated whether the 3-gene signature could provide a solution to forward the window of NAAT tests. Note that the E-MTAB-10022 consisted of asymptomatic samples collected before the timepoint of the first SARS-CoV-2 PCR positivity (referred to as “PCR positive -1”). Thereafter, this 3-gene signature could identify the SARS-CoV-2 infections before detectable viral RNA on RT-PCR assay (Figure 6B).

### 3.6. Comparisons of Gene Expression Signatures

Multiple studies have presented gene expression signatures that discriminate between bacterial and viral infections according to various computational strategies [12,13,14,15,16]. To compare with other signatures including similar sizes of genes, we systematically enrolled signatures on the following conditions: (1) the size of genes must be less than five; (2) genes in the signature must be available in the discovery train dataset; four out of six signatures were kept (Appendix A). For each signature, we calculated the AUC of all SARS-CoV-2 cohorts, with the score of signatures in the case group being larger than that in the control group. As shown in Figure 7A, our 3-gene signature outperformed others in discriminating asymptomatic SARS-CoV-2 infections, especially before the positivity of PCR tests. In addition, the 3-gene signature ranked first in diagnostic power among HRA000786, GSE152641, GSE161918, GSE171110, GSE157103, and GSE201530. Meanwhile, the 3-gene signature ranked second in GSE198449, E-MTAB-10022, weaker than TS1 (Figure 7B). Collectively, the 3-gene signature demonstrated an overall better performance in each SARS-CoV-2 cohort than almost all studied signatures.

### 3.7. Query a Gene of Interest via ViRAL Online

To facilitate the re-use of data collected in this study, we constructed ViRAL (Viral Immune Response Analytics), a web-based tool to deliver a suite of advanced functions for the characterization of a gene of interest across viral infections (Figure 8). Users can query a gene of interest by typing the gene symbol in the “Search a Gene” field or selecting the gene from the drop-down list. ViRAL provided: (i) Pan-respiratory-virus differential expression (DE) analysis to determine if the gene of interest is differentially expressed between infected and un-infected samples presenting symptoms or not, and ROC analysis is performed to measure the performance of the gene biomarker in distinguishing infected samples from control samples with a forest plot, (ii) The DE and ROC analyses can also be implemented for a selected single or multiple viral infection project to show more detailed information about the dynamics of the gene and its associated diagnostic power for a viral infection type of interest. (iii) To address the question “What cells were the major sources of the gene?”, UMAP projection for cell identity and expression of the gene of interest from scRNA-seq were presented, along with the violin plot with gene expression per cell in the group of infected and uninfected samples.

## 4. Discussion

Research on the host responses to respiratory viruses could help develop effective interventions and therapies against the current and future pandemics from the host perspective. Here, we leveraged the substantial biological, clinical, and technical heterogeneity in publicly available respiratory virus datasets and identified a 3-gene (CLSPN, RBBP6, and CCDC91) host response signature for identifying the SARS-CoV-2 infection at either an asymptomatic or symptomatic stage. We validated this 3-gene signature in an independent longitudinal follow-up study and demonstrated that it performs well (AUC > 0.9), even before the development of detectable viral RNA on RT-PCR testing. Our discovery and validation cohorts were from the continents of Asia, Europe, and North America, providing strong evidence that the 3-gene signature is not undermined by the underlying genetic background of the patients or the virus strains, showing potential for translation to clinical practice.

Host responses resulting in SARS-CoV-2 infection differed from several common respiratory viruses. These results were in line with previous in vitro studies [9] (Appendix A). Although part of ISGs (i.e., IFI27) were enhanced in SARS-CoV-2 infection (Figure 2A), moderate IFN responses have been a sign of COVID-19 (Figure 2C). The improved cellular immune response, such as strong T cell responses and high secretion of TNF, were observed, surprisingly in these asymptomatic subjects (Figure 2 and Figure 3). It has been reported that asymptomatic SARS-CoV-2-infected individuals might develop an efficient antiviral cellular immunity to protect the host without causing any apparent symptoms [51,52]. In addition, the results of CIBERSORTx revealed a reduced level of monocytes in SARS-CoV-2 infection, while other viruses introduced an elevated level of monocytes, especially in the symptomatic group (Figure 3A). SARS-CoV-2-related monocytic abnormalities have been observed in both symptomatic [53] and asymptomatic cases [17]. The possible mechanism might be that the monocytes could be directly infected with SARS-CoV-2, leading to pyroptosis [54]. Moreover, attention should be paid to the boosted levels of IFNG and TNF in COVID-19 (Figure 3B), as synergism of TNF-α and IFNG triggers inflammatory cell death, tissue damage, and mortality in SARS-CoV-2 infection [55].

Discovery and validation of SARS-CoV-2-specific host response genes have been calling [26]. To address this issue, an ideal dataset for biomarker discovery should include not only SARS-CoV-2 infections but also other respiratory infections (Figure 4). Therefore, we co-normalized datasets HRA000786 and GSE17156, which included both asymptomatic and symptomatic cases. With integrative bioinformatics and machine learning approaches, we identified that the combination of CLSPN, RBBP6, and CCDC91 was robustly associated with SARS-CoV-2 infection in both discovery and validation datasets. Collectively, the specificity of these genes in the context of SARS-CoV-2, as compared to other respiratory viruses, suggests distinct molecular mechanisms that may be pivotal in the pathogenesis.

We proposed that the upregulation of CLSPN, RBBP6, and CCDC91 by SARS-CoV-2 in T cells might create a cascade of effects that alter both the innate and adaptive immune responses. First, CLSPN plays a significant role in genomic stability during DNA replication [56]. During viral infections, the expression of CLSPN could be upregulated in response to DNA damage induced by viral replication processes [57]. Given that T cells need to rapidly proliferate and function efficiently due to SARS-CoV-2 infection, CLSPN might help T cells manage the stress and damage induced by viral replication. Second, known for its interaction with p53 and retinoblastoma protein (RB), RBBP6 is involved in cell cycle regulation, apoptosis, and ubiquitination processes [58]. RBBP6 has been reported as a negative regulator of Ebola virus replication by mimicking the viral protein [59]. The specific upregulation of RBBP6 during SARS-CoV-2 infection might indicate a defensive host response aimed at curtailing viral replication through these interactions. Moreover, the role of RBBP6 in T cells could regulate their lifecycle. RBBP6 might prevent premature T cell death during the active response phase and ensure appropriate apoptosis afterward to avoid autoimmunity or chronic inflammation. Third, CCDC91 is involved in the regulation of membrane traffic through the trans-Golgi network. The modulation of CCDC91 expression in SARS-CoV-2 infections might reflect the virus’s strategy to alter host intracellular trafficking routes for its benefit, promoting viral assembly and egress. Although there is limited specific information on the expression changes of CCDC91 in response to viral infections, CCDC50, a related gene to CCDC91, negatively regulated the type I IFN signaling pathway initiated by RIG-I-like receptors (RLRs), the sensors for RNA viruses [60]. Together, these genes may in concert with T cells not only respond to viral infections effectively through cellular immunity but also regulate the immune response to avoid excessive inflammation or autoimmunity. Further experimental validation and research would be essential to elucidate these proposed mechanisms fully and to understand how these genes specifically affect T cell dynamics in the context of SARS-CoV-2.

The 3-gene signature offered two major improvements to SARS-CoV-2 infection triage: specificity and early diagnostic power. External datasets without co-normalization were collected for validation, and the 3-gene signature showed superior diagnostic performance only in SARS-CoV-2 (Figure 6A). Importantly, the 3-gene signature achieved excellent diagnostic performance in Omicron patients (GSE201530). Additionally, the 3-gene signature showed exceptional performance to identify SARS-CoV-2 infection before detectable viral RNA on RT-PCR testing with an AUC of 0.940 (95% CI 0.875–1) (Figure 6B).

The 3-gene signature outperformed in the identification of SARS-CoV-2 than published signatures composed of various genes (Figure 7). First, published signatures without the generalization ability in SARS-CoV-2 may arise from each signature that was not developed for SARS-CoV-2 infection. The 3-gene signature here was dimensionally reduced by integrative bioinformatics and machine learning approaches. The bioinformatics methods, including DEGs and WGCNA, provided biological insights into the stable SARS-CoV-2 specific candidate genes. The machine learning approaches on these candidates offered a better extrapolation possibility. Second, the genes in these public signatures were associated with interferon activities, a driving feature of common respiratory viral infections, but not COVID-19. Third, a previous study suggested that the host response to viral infection was dominated by myeloid cells [37]. Noting that the T cells, which are the primary sources of the 3-gene signature, might contribute to effectiveness, which requires future studies. The SARS-CoV-2-specific 3-gene signature not only offers a diagnostic tool but also enhances our understanding of the biological processes most disrupted by SARS-CoV-2, potentially guiding targeted therapies.

Novel and user-friendly tools are desperately needed to facilitate data mining of a wealth of publicly available gene expression data. ViRAL enables experimental virologists without any computational programming skills to perform a diverse range of gene expression analyses over a gene of interest. By using ViRAL, experimental biologists can easily ask specific questions and test their hypotheses. For example, one can easily find the gene CLSPN, which is specifically expressed in SARS-CoV-2 infection regardless of symptom (Figure 8).

Nevertheless, our study had certain limitations. First, the co-normalization of discovery train datasets may lose some biological signals. However, the 3-gene signature was validated in all the datasets without co-normalization which in turn showed support that the host response to viral infections was robust across cohorts. Second, limited types of viral infections were validated. Although the IFV, RSV, and HRV were the three leading viral pathogens [7], other popular viruses, e.g., human parainfluenza virus (HPIV) and human adenoviruses (HAdV), would need to be tested in the future. Third, critical considerations for the clinical application of the 3-gene signature were the temporal and clinical resolution of the test. The incubation time of Omicron was shortened to 3.49 days [61], compared to the 5–7 days for the earlier strains. Given that the 3-gene signature could identify Omicron patients (Figure 6), the signature would be worth future longitudinal studies to assess diagnostic performance in different Omicron variants. In addition, none of the studies included in the analysis provided information on comorbidities. Hence, we could not assess how the 3-gene signature would perform in patients with comorbidities, especially immune system-related disorders. Future studies validating the 3-gene signature should focus on addressing these limitations.

In summary, our study aimed to uncover the unique pathogenesis of COVID-19 by investigating the differential host responses to prevalent viruses. We systematically identified a SARS-CoV-2-specific 3-gene signature that could independently distinguish asymptomatic and symptomatic individuals infected with SARS-CoV-2 from other respiratory viral infections. Given that T cells were the major source of the 3-gene signature, the importance of T cells in combating COVID-19 needed to be further studied. Overall, the development of tools through host responses to viruses could serve as a promising tool to combat future viral pandemics.

## Figures and Tables

**Figure 1 viruses-16-01029-f001:**
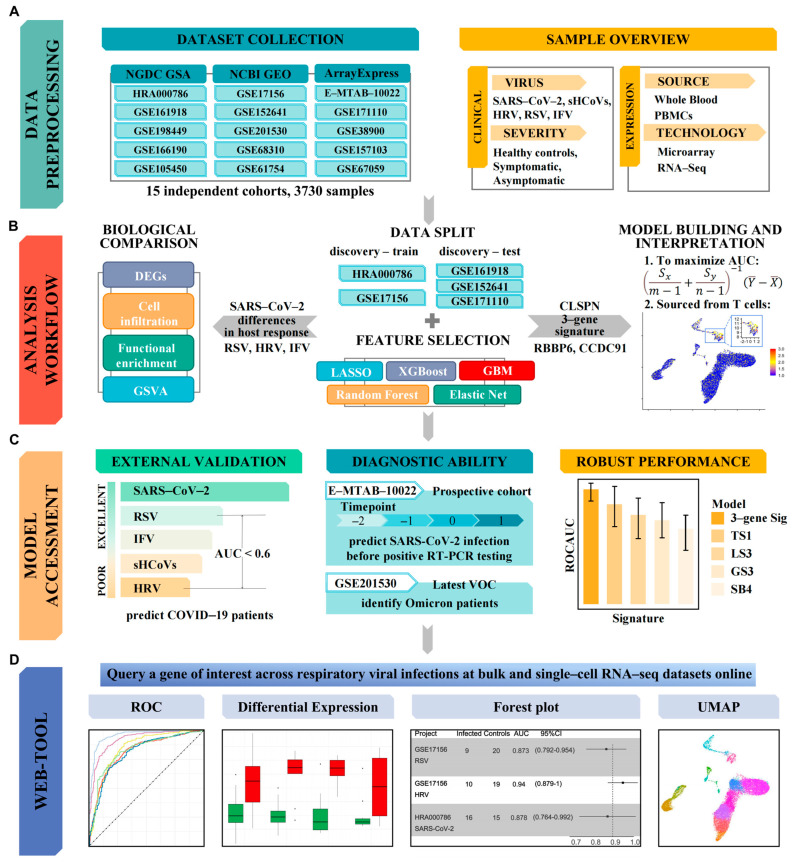
The overall workflow of this study. (**A**) Data collection, curation, and preprocessing. (**B**) Profiling of host response to viral infections and identification of SARS-CoV-2 specific signature. (**C**) Predictive value and robust performance. (**D**) Querying a gene of interest via ViRAL online.

**Figure 2 viruses-16-01029-f002:**
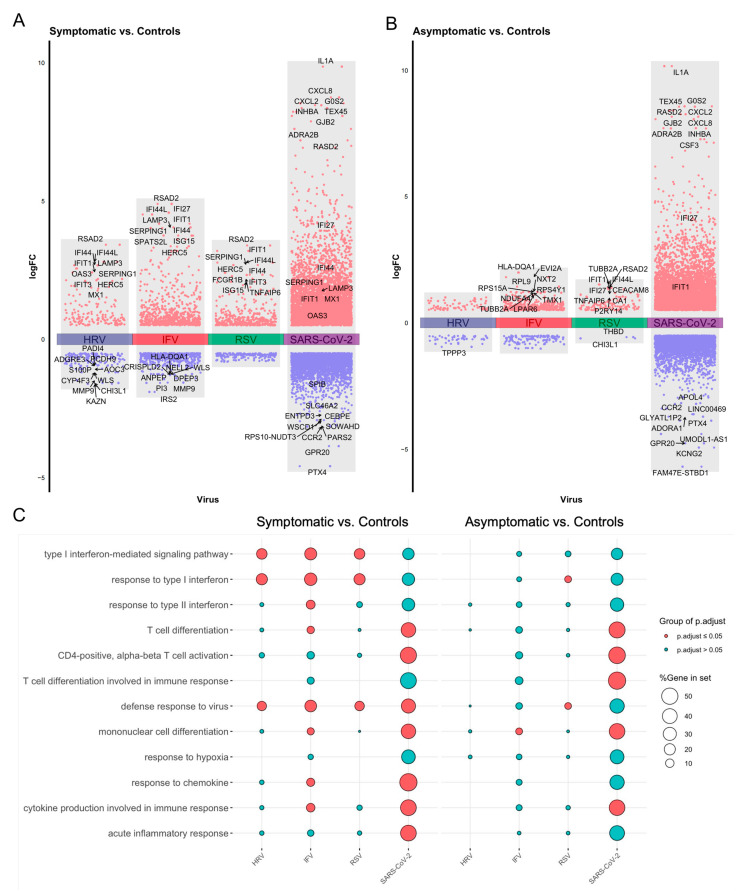
Differential expression genes (DEGs) and gene ontology (GO) analysis of asymptomatic and symptomatic cases with viral infections. DEGs in contrasting (**A**) asymptomatic and (**B**) symptomatic against healthy controls in each cohort. (**C**) Enrichment results of GO analysis via DEGs derived from viral infections, respectively.

**Figure 3 viruses-16-01029-f003:**
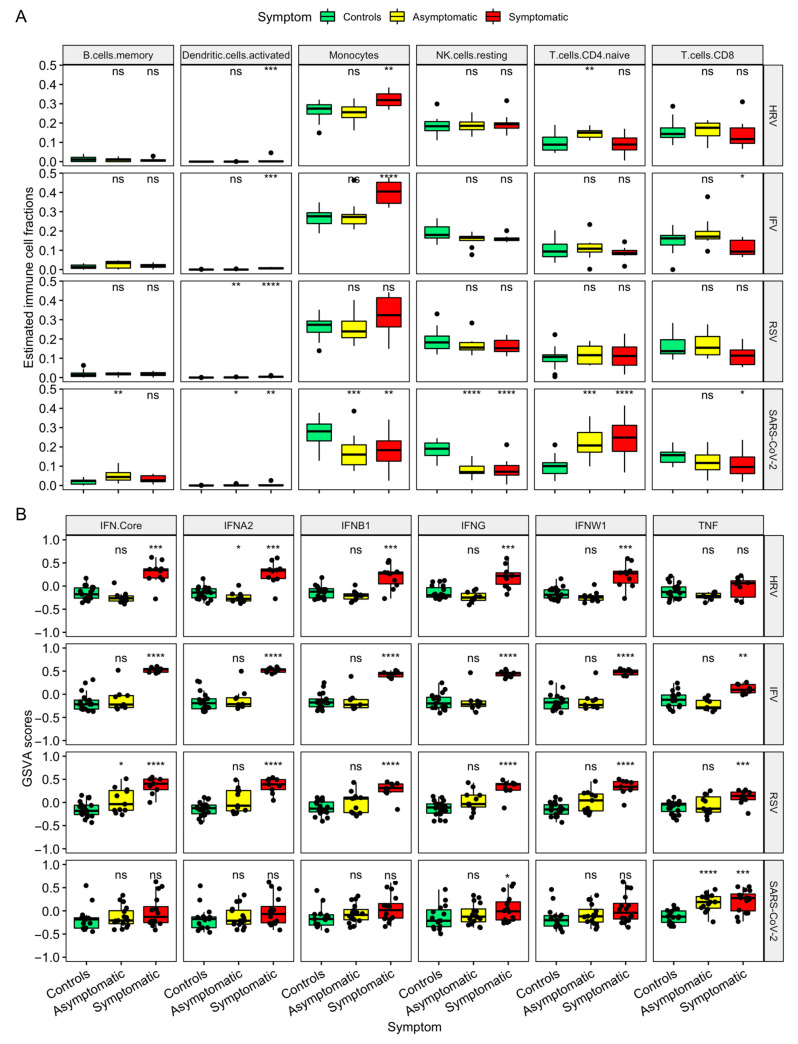
Host response landscape of SARS-CoV-2 and common respiratory viruses. (**A**) Estimated immune cell fractions using CIBERSORTx through the discovery train set, HRA000786 and GSE17156. (**B**) Interferon subtype signatures based on GSVA scores. The GSVA scores were ranging from −1 to 1. A positive GSVA score means that the pre-defined gene set in a sample has a greater expression than the same gene set with a negative value. GSVA: Gene Set Variation Analysis (ns (not significant) * *p* ≥ 0.05, ** *p* ≤ 0.01, *** *p* ≤ 0.001, **** *p* ≤ 0.0001).

**Figure 4 viruses-16-01029-f004:**
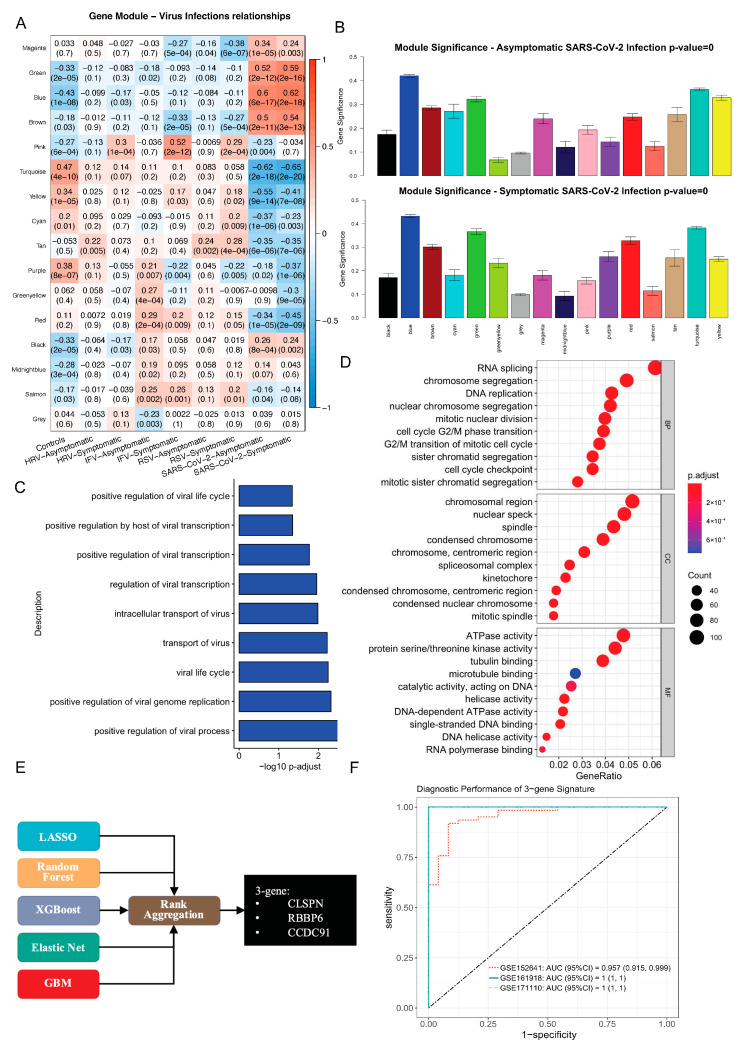
Integrative bioinformatics and machine learning approaches uncovered SARS-CoV-2 infection-specific feature genes. (**A**) Heatmap of the correlation between module eigengenes and viral infections. (**B**) Distribution of average gene significance and errors in the modules associated with asymptomatic and symptomatic SARS-CoV-2 infection. The virus-related (**C**) and most enriched (**D**) GO results. (**E**) The SARS-CoV-2 specific genes, CLSPN, RBBP6, and CCDC9, were discovered by robust rank aggregation through genes selected by five machine learning methods. (**F**) Performance of the 3-gene signature in discovery test datasets.

**Figure 5 viruses-16-01029-f005:**
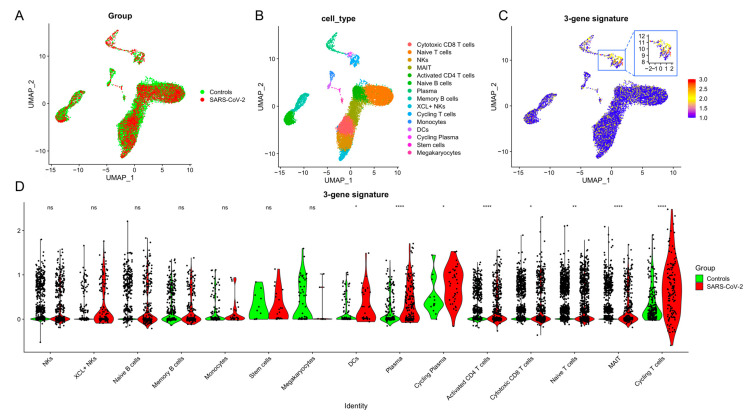
The 3-gene signature for SARS-CoV-2 infection was predominantly expressed by T cells. UMAP representation of scRNA-seq data from CNP0001102. A–C represent clustering based on clinical status (**A**), cell type (**B**), and 3-gene signature (**C**). (**D**) The violin plot shows 3-gene signature distribution in cell types by clinical status. The *p*-values were calculated using Mann-Whitney U test (ns (not significant) * *p* ≥ 0.05, ** *p* ≤ 0.01, **** *p* ≤ 0.0001).

**Figure 6 viruses-16-01029-f006:**
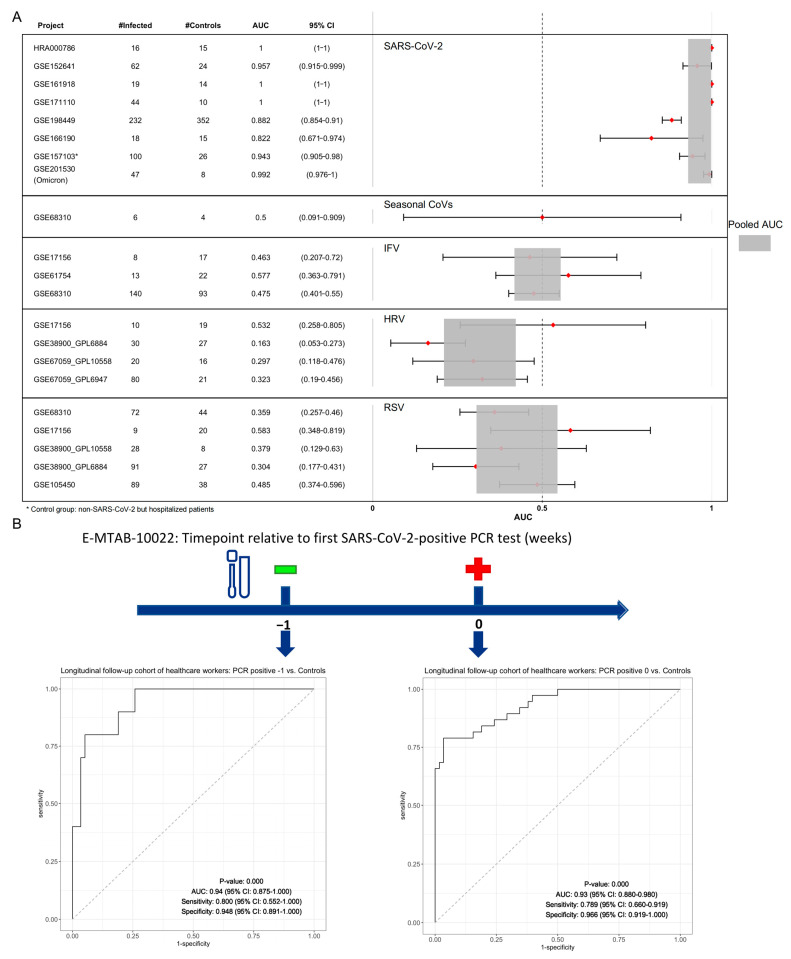
Robust performance and predictive value of the SARS-CoV-2 specific 3-gene signature. (**A**) Forest plot of diagnostic performance of the 3-gene signature on discovery and validation cohorts without co-normalization. (**B**) The 3-gene signature could predict the positivity of the SARS-CoV-2 PCR test in a prospective cohort.

**Figure 7 viruses-16-01029-f007:**
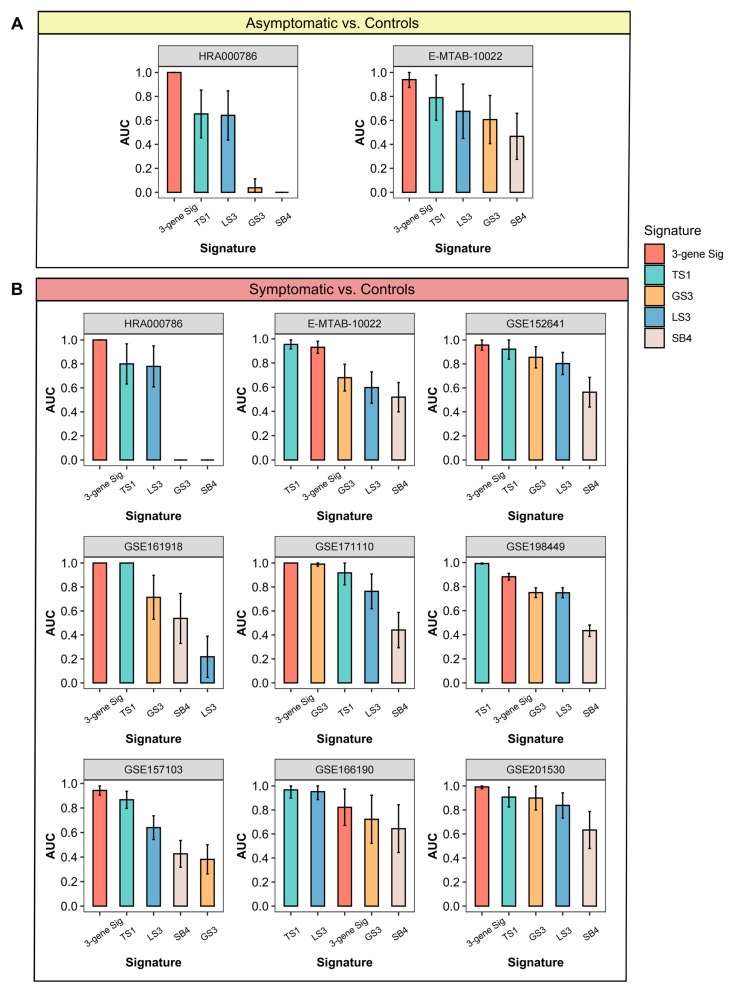
Comparisons of gene expression signatures. AUC of the 3-gene signature and four published signatures in all SARS-CoV-2 cohorts without co-normalization in asymptomatic (**A**) and symptomatic (**B**) groups.

**Figure 8 viruses-16-01029-f008:**
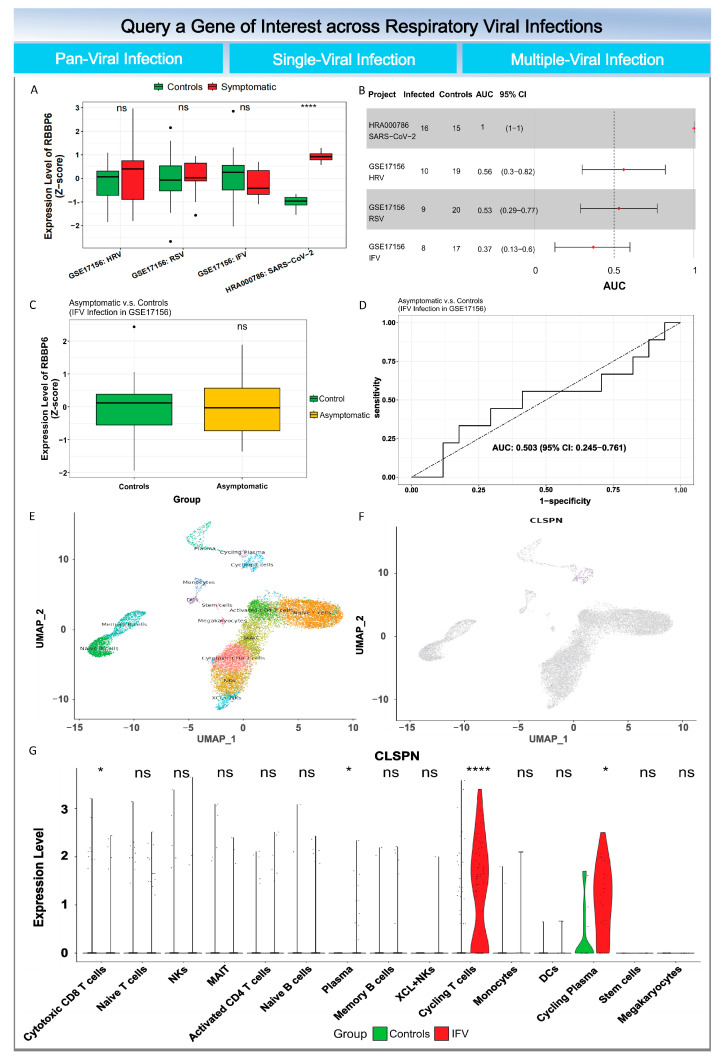
Outputs of ViRAL from the query of a gene of interest. (**A**) Pan-respiratory-virus differential expression analysis across discovery datasets. (**B**) A forest plot visualizing pan-respiratory-virus AUC analysis across discovery datasets. (**C**) Boxplot of the gene expression in infected and uninfected samples from the selected viral infection project. (**D**) A ROC curve illustrating the diagnostic ability of the gene in the selected project. UMAP projection for cell identity (**E**) and expression of gene of interest (**F**). (**G**) Violin plot with gene expression per cell in the group of infected and normal samples. AUC, area under curve; ROC, receiver operating characteristic. ns (not significant), * *p* ≥ 0.05, **** *p* ≤ 0.0001.

## Data Availability

The datasets studied can be found through the accession number as shown in this study. ViRAL is publicly available at http://viral.org.cn:3838/ViRAL (accessed on 23 July 2023). The source code of ViRAL is available at https://github.com/yeli7068/ViRAL (accessed on 23 July 2023).

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
