# Peer review of "A T-Cell-Derived 3-Gene Signature Distinguishes SARS-CoV-2 from Common Respiratory Viruses"

_viruses, 2024, doi:10.3390/v16071029_

Round 1

Reviewer 1 Report

Comments and Suggestions for Authors

1) Will be nice to explain figure 1 in legend as legend is missing in figure 1

2) Whether information related to PCR results for samples collected or used for study was present should be mentioned in draft.

3) Information related to different variants of SARS-CoV-2 in samples used for study should be included

4) Improve the quality of images (fig 4)

5) Text in figures are sometime difficult to read, authors are asked to fix this issue throughout the manuscript

Comments on the Quality of English Language

Minor editing is needed

Author Response

Response to reviewer, 

Thanks for your valuable feedback and draw our attention to several important details. We apologized for any inconvenience and confusion caused by the quality of pictures. After a throughout reviewing, we have carefully addressed each of your comments and made the necessary revisions, which are outlined below. We expect these adjustments have strengthened our manuscript provide a more comprehensive understanding of our study. Our response is given in normal font and changes/additions to the manuscript are given in the red text.

Comment 1: Will be nice to explain figure 1 in legend as legend is missing in figure 1

Reply: Thanks for bringing this oversight to our attention. To improve the clarity and readability of Figure 1, we have reorganized it into four sub-figures, each with its own legend that provides a brief description of the workflow:

Figure 1. The overall workflow of this study. (A) Data collection, curation, and preprocessing. (B) Profiling of host response to viral infections and identification of SARS-CoV-2 specific signature. (C) Predictive value and robust performance of SARS-CoV-2 specific signature. (D) Querying a gene of interest via ViRAL online.

Comment 2: Whether information related to PCR results for samples collected or used for study was present should be mentioned in draft.

Reply: Thanks for pointing this out. In this research, both the discovery and validation cohorts were obtained from the public dataset. The infection status of the samples was determined based on the results from established techniques for virus detection, such as viral culture and polymerase chain reaction (PCR). The details have been described in the original publications. Modifications were shown in lines 101-104.

Comment 3: Information related to different variants of SARS-CoV-2 in samples used for study should be included.

Reply: Thanks for emphasizing the importance of providing details on the different variants of SARS-CoV-2. While we understand the significance of variant information, we must acknowledge that not all datasets provide this level of detail. Most datasets only confirm the presence or absence of SARS-CoV-2, without further specifying the variant. For those datasets that do include variant information, we have provided the variants information in the Table S1.

Comment 4: Improve the quality of images (fig 4)

Reply: Sorry for the inconvenience. All the figures were replaced with a high-resolution version.

Comment 5: Texts in figures are sometime difficult to read, authors are asked to fix this issue throughout the manuscript

Reply: To address this issue, we have re-edited the figures, reformatted the text, and optimized the contrast to ensure a better quality.

Reviewer 2 Report

Comments and Suggestions for Authors

The manuscript is well-written and presents significant findings from an in-depth analysis of public datasets on patients with respiratory viruses. The amount of data considered in this study is adequate, and there should be no issues regarding ethics committee permissions, although this should be explicitly stated in the manuscript.

However, several issues need to be addressed:

- The introduction (or the discussion) should better describe the other signatures used for comparison in the study, emphasizing the biological relevance of the selected genes.

- The discussion should delve into the molecular mechanisms by which the selected 3-gene signature is specifically involved in SARS-CoV-2 infection. The current explanation is somewhat reductive. The authors should elaborate on the functions of the selected genes, including when their expression has been reported to be up- or down-regulated, and propose mechanisms based on both literature data and their hypotheses.

- Line 449: "in vitro studies" should be briefly described, with appropriate references added, as reference 46 seems inappropriate. Additionally, an experimental validation with an in vitro experiment would further substantiate the data, or alternatively, the use of databases from in vitro studies could serve this purpose.

- The possible comorbidities and the lack of data stratification based on the immune status of patients is a significant limitation that could bias the analysis. The authors should demonstrate that this aspect does not invalidate the study by providing appropriate proof of concept.

- The ViRAL link is not functional and should be corrected.

- The quality of the figures is too low, and the small writing is unclear.

- Formulas should be moved to the materials and methods section.

Author Response

Response to Reviewer,

We feel great thanks for your professional review work on our article. As you are concerned, there are several issues that need to be addressed. According to your nice suggestions, we have made extensive corrections to our previous draft, the detailed corrections are listed below and marked in red in the manuscript.

Comment 1: The manuscript is well-written and presents significant findings from an in-depth analysis of public datasets on patients with respiratory viruses. The amount of data considered in this study is adequate, and there should be no issues regarding ethics committee permissions, although this should be explicitly stated in the manuscript. To ensure the manuscript is complete and transparent about the ethical considerations, a section on ethics should be explicitly included. Here’s a suggested addition that could be integrated into the manuscript to address this aspect:

Reply: Thanks for pointing this out. We have added a section Ethics approval as shown in line 583-587: The present study utilized publicly available data and did not involve the recruitment of any new human subjects. The data used was previously collected as described by the original studies and is fully anonymized. Our research approach and methodology adhered strictly to the principles of the Declaration of Helsinki as they pertain to the ethical conduct of research involving human subjects.

Comment 2: The introduction (or the discussion) should better describe the other signatures used for comparison in the study, emphasizing the biological relevance of the selected genes.

Reply: We think this is an excellent suggestion. We have added the contexts about the published signatures in both introduction and discussion as shown in lines 71-76 for introduction and lines 522-536 for discussion.

The introduction added a new paragraph which described the purposes of these signatures, including discrimination of bacterial/viral infection, diagnosis of pre-symptomatic viral infection and identification of specific viral infection.

We re-wrote a part of discussion. The part mainly described the why the SARS-CoV-2 specific 3-gene signatureoutperformed the published signatures in differentiating SARS-CoV-2 from other respiratory viral infections. Briefly, the 3-gene signature were developed for discriminating SARS-CoV-2 from other respiratory viruses while others focus on discriminating viral from bacterial infection or no infection. Next, the genes included by published signature were mainly interferon-induced genes. Noting that impaired interferon activities have been considered as a feature of COVID-19. Finally, although the host response to viruses were typically dominated by myeloid cells, the proposed SARS-CoV-2 specific 3-gene signature were mainly expressed by T cells here, suggesting the cellular response played important roles in the antiviral activities during SARS-CoV-2 infections.

Comment 3: The discussion should delve into the molecular mechanisms by which the selected 3-gene signature is specifically involved in SARS-CoV-2 infection. The current explanation is somewhat reductive. The authors should elaborate on the functions of the selected genes, including when their expression has been reported to be up- or down-regulated, and propose mechanisms based on both literature data and their hypotheses.

Reply: We agree with the reviewer that further elaborating on this point could be helpful. The functions of these genes,CLSPN, RBBP6, and CCDC91, were carefully investigated on their expression in the context of viral infections as shown in lines 487 to 513.

Generally, the 3-gene work in concert within T cells might not only respond to viral infections effectively through cellular immunity but also regulate the immune response to avoid excessive inflammation or autoimmunity. First, CLSPN played a significant role in genomic stability during DNA replication. During viral infections, the expression of CLSPN could be upregulated in response to DNA damage induced by viral replication processes. Given that T cells need to rapidly proliferate and function efficiently due to SARS-CoV-2 infection, CLSPN might help T cells manage the stress and damage induced by viral replication. Second, known for its interaction with p53 and retinoblastoma protein (RB), RBBP6 is involved in cell cycle regulation, apoptosis, and ubiquitination processes. RBBP6 has been reported as a negative regulator of Ebola virus replication via mimicking the viral protein. The specific upregulation of RBBP6 during SARS-CoV-2 infection might indicate a defensive host response aimed at curtailing viral replication through these interactions. Moreover, the role of RBBP6 in T cells could regulate their lifecycle. RBBP6 might prevent premature T cell death during the active response phase and ensuring appropriate apoptosis afterwards to avoid autoimmunity or chronic inflammation. Third, CCDC91 is involved in the regulation of membrane traffic through the trans-Golgi network. The modulation of CCDC91 expression in SARS-CoV-2 infections might reflect the virus's strategy to alter host intracellular trafficking routes for its benefit, promoting viral assembly and egress. Although there is limited specific information on the expression changes of CCDC91 in response to viral infections, CCDC50, a related gene to CCDC91, negatively regulated the type I IFN signaling pathway initiated by RIG-I-like receptors (RLRs), the sensors for RNA viruses. Further experimental validation and research would be essential to elucidate these proposed mechanisms fully and to understand how these genes specifically affect T cell dynamics in the context of SARS-CoV-2.

Comment 4: Line 449: "in vitro studies" should be briefly described, with appropriate references added, as reference 46 seems inappropriate. Additionally, an experimental validation with an in vitro experiment would further substantiate the data, or alternatively, the use of databases from in vitro studies could serve this purpose.

Reply: We sincerely thank the reviewer for careful reading. We have corrected to the reference “Blanco-Melo D, Nilsson-Payant BE, Liu W-C, Uhl S, Hoagland D, Møller R, et al. Imbalanced Host Response to SARS-CoV-2 Drives Development of COVID-19. Cell. 2020;181:1036-1045.e9.”

To better illustrate the host responses result in SARS-CoV-2 infection differed from several common respiratory viruses with in vivo and in vitro studies, we performed Differential expression genes (DEGs) analysis of A549 cells with different viral infections as shown in Figure S5. The antiviral interferon-stimulated genes (ISGs) were induced by respiratory viruses other than SARS-CoV-2 in A549 cells, suggesting the similar interferon activities induced by SARS-CoV-2 in both in vivo and in vitro studies.

Comment 5: The possible comorbidities and the lack of data stratification based on the immune status of patients is a significant limitation that could bias the analysis. The authors should demonstrate that this aspect does not invalidate the study by providing appropriate proof of concept.

Reply: It is true that the comorbidities and varying immune statuses in patients is crucial for substantiating the reliability of the study's findings. However, datasets with immune statuses were not feasible so far. We admit this is a limitation of this study as shown in lines 555-562. We hope, in the future, a prospective cohort study designed to track the transcriptomics in patients with and without comorbidities upon exposure to SARS-CoV-2 and other viruses could provide direct evidence of the signature’s utility. Such a study would help establish whether changes in the expression of these genes are consistent predictors of SARS-CoV-2 infection across varying immune statuses and health backgrounds.

Comment 6: The ViRAL link is not functional and should be corrected.

Reply: Thanks for using ViRAL. The link of ViRAL has been reset and tested by peers either in- or outside- China. We also asked the IT department to provide helps to maintain the access. Besides, the source codes of ViRAL are available athttps://github.com/yeli7068/ViRAL.

Comment 7: The quality of the figures is too low, and the small writing is unclear.

Reply: Sorry for the inconvenience. All the figures were replaced with a high-resolution version.

Comment 8: Formulas should be moved to the materials and methods section.

Reply: Thanks for the suggestion. We moved the formulas to method section as shown in lines 225-228.
